# Research Trends in Green Product for Environment: A Bibliometric Perspective

**DOI:** 10.3390/ijerph17228469

**Published:** 2020-11-16

**Authors:** Amit Kumar Bhardwaj, Arunesh Garg, Shri Ram, Yuvraj Gajpal, Chengsi Zheng

**Affiliations:** 1L.M. Thapar School of Management (Dera Bassi Campus), Thapar Institute of Engineering & Technology, Patiala 147004, India; akbhardwaj@thapar.edu (A.K.B.); arunesh.garg@thapar.edu (A.G.); shriram@thapar.edu (S.R.); 2Asper School of Business, University of Manitoba, Winnipeg, MB R3T 5V4, Canada; yuvraj.gajpal@umanitoba.ca; 3School of Finance and Trade, Wenzhou Business College, Wenzhou 325035, China

**Keywords:** sustainability, sustainable development, green marketing, green product, literature review, bibliometric analysis

## Abstract

The term “green products” is used commonly to describe the products that seek to protect or enhance the environment during production, use, or disposal by conserving resources and minimizing the use of toxic agents, pollution, and waste. Hence, green products offer potential benefits to the environment and human health. Therefore, environmentally conscious consumers have shown an enhanced inclination for them. Consumer preferences, environmental activism, and stringent regulations have forced sustainability-oriented firms to shift their focus to producing green products. The present study uses bibliometric tools and various indicators to discern research progress in the field of green products over the period 1964–2019. Further, VOSviewer software is applied to map the main trends. A total of 1619 publications during the study period were extracted from the SCOPUS database using different keywords related to the green products. The data analysis indicates that the field of green products has experienced significant growth since 1964, especially in the last 14 years. In terms of publications and citations, the United States is the leading country. The field of research concerning green products has evolved from the early debates on sustainable design, green marketing, sustainable development, and sustainability. The topic seems to be advancing into a variety of green themes related to consumer trust and purchase intentions, branding and loyalty, and environmental and health consciousness.

## 1. Introduction

Over the last few decades, there has been a worldwide realization of the importance of preserving environmental health. Many available studies have drawn attention to the environmental risks associated with growing consumerism and industrial production [1,2,3]. The increased industrial output has been held responsible to a large extent for many negative environmental impacts including loss of natural resources, air and water pollution, climate change leading to global warming, life-threatening diseases, and extinction of species [4,5]. Therefore, responsible behavior by society and business firms is vital to achieving environmental sustainability in the future [6]. Environmental sustainability entered the agenda of policymakers in many countries after the first United Nations Conference on the Human Environment held in Stockholm in 1972 [7]. However, sustainability as a business practice gained prominence after the appearance of the concept of sustainable development in 1987, which emphasized the need for human development along with environmental protection [6]. Ever since then, the sustainability concerns related to the integration of environmental and economic goals have become the forethought of the leading business firms, and they are increasingly adopting green practices. “Green”, here, is tending to take care of the environment and improve its quality.

To address environmental sustainability, the firms have generally focused on clean technologies and pollution prevention by adopting two kinds of environmental strategies, viz., process-oriented and organization-oriented [1]. Process-oriented environmental strategies are focused on clean technologies including cleaner production, material eco-efficiency, material saving, renewable energy technologies, and efficient energy utilization [8,9,10,11]. Organization-oriented strategies include extending environmental strategies to the supply chain [12].

Since the early 2000s, the focus of sustainability-oriented firms has shifted from the adoption of clean technologies to producing environment-friendly green products [13]. This shift may be attributed to the fact that green products strive to protect or enhance the environment by conserving energy and/or resources and reducing or eliminating the use of toxic agents, pollution, and waste [14]. Green products offer high quality and low overall costs to the consumer and society as these products are characterized by efficient use of resources and low risks to the environment since the inception phase [1]. The stringent environmental regulations aimed at minimizing the ecological footprint of products, environmental activism, and pressure exerted by environmental lobbies [15] are further responsible for the interest of firms in delivering green products. However, the augmented concern for the environment among consumers, known as green consumerism, is considered to be majorly instrumental in influencing firms to produce green products [16]. The environmentally conscious consumers tend to shift their consumption preferences for environment-friendly green products [17]. Hence, a new business opportunity for firms in the form of a market for green products has emerged [18]. “Green products” as a term has been coined largely within the marketing field, and its popularity has coincided with the environmental awakening of the consumer [19]. Thus, the field of green products holds special relevance to the domain of Marketing Management.

The other similar terms used in the literature for green products include environmental products, ecological products, eco-product, and sustainable products. There are more than 50 definitions of green products [19]. Researchers have linked green products with environment compatibility, environment protection, environment friendliness, environment sustainability, reduced wastage during production, environment-friendly production, social quality, ethical attributes, economic benefits, durability, recyclability, resource-conservation potential, toxic-free ingredients, low energy consumption, low emissions, less packaging, protection of public health, etc. [14,16,20,21,22,23,24,25,26,27]. Further, there is a lack of convergence of the viewpoints of academicians, businesses, and consumers on what constitutes a green product [28]. Hence, the term is open for interpretation and debate [29] primarily due to the lack of a commonly accepted definition [19]. This ambiguity offers scope for researchers to examine the concept of green products and related aspects and contribute to its understanding.

The relevant literature indicates that existing studies have undertaken systematic reviews for green product innovation [7], green product development [30], green product and process innovation [31], sustainable product innovation [32], eco-innovation [33], and sustainability-oriented innovation [34]. However, very few studies [19,28] have dealt with green products. The available reviews have put forth a variety of topical findings offered by the existing studies. Systematic reviews of a body of literature in a particular area identify knowledge outcomes and paradigm shifts [35]. However, further analysis of the available literature using bibliometric tools may offer additional insights into the growth of the concept of green products. The bibliometric tools enable longitudinal analysis of the literature, illustrate the evolution of the topic, discern key clusters of research within the topic, and identify current and potential areas of research [36].

The available studies reveal that researchers have used bibliometric tools to examine the existing literature related to various green and sustainable offerings, processes, and practices like green manufacturing [37], green supply chain [36,38], green innovation [39], green building [40], green energy [41], sustainable construction [42], sustainable design [43], and sustainable manufacturing [44]. However, hardly any study has attempted to undertake a bibliometric analysis of the literature in the field of green products. Therefore, to fill this key research gap, the present study uses bibliometric tools to systematically examine how the research concerning the field of green products has progressed over time. Since the concept of green products has majorly evolved in the domain of Marketing Management [19], the present study has focused on keyword identification and research areas relevant to green products in the domain of Marketing Management.

The major contribution of the present study is a bibliometric analysis of the research concerning green products and related aspects over the years 1964–2019. The outcome of the study includes the publication and citation growth pattern; most prominent papers; leading authors, journals, institutions, and countries; and network analysis of the co-occurrence of the author-supplied keywords concerning the green products. The findings will offer insights for researchers and readers to identify the evolution and growth of the topic of green products and ascertain current and potential areas of research.

The rest of this study is structured as follows. The next section describes the research methods. The third section presents the results of the bibliometric analysis. This section is followed by a discussion of the results. Finally, the fifth section concludes the study with limitations and directions for future research.

## 2. Materials and Methods

To achieve the objective of a study using bibliometric tools, it is necessary to identify, collect, classify, and consolidate the available published knowledge on the chosen topic and related aspects. For the purpose, it is required to follow an iterative cycle of defining appropriate search keywords, scanning the available resources in the literature, collecting and organizing the relevant data, and carrying out the further analysis using relevant bibliometric tools as suggested by the existing studies [45,46]. The present study has employed a similar approach for data collection and evaluation of the literature concerning green products and related aspects to identify the key research work carried out in this area over the period of 1964–2019. The collected data has been further analyzed by using VOSViewer (Visualization of Similarities Viewer, created at Leiden University, Leiden, the Netherlands) software for bibliometric analysis to offer insights into the current research interests and directions for future research in the field of green products. This is one of the most commonly used open-source tools and offers network visualization of authors, institutions, and keywords and their association through cluster analysis [47,48]. The internationally widely used free bibliometric analysis software VOSViewer has been applied by many bibliometric studies [49,50,51] in the domain of management.

For the purpose of the study, a multifold approach for data collection and analysis as represented in Figure 1 has been used. This included keyword identification; literature search; removal of duplicate records; data cleansing considering language, document type, and research areas; bibliometric analysis using Microsoft Excel and VOSViewer software; and further analysis to identify the research trends in the field of green products.

A list of 60 keyword phrases (shown in Table 1) relevant to the field of green products in the domain of Marketing Management was first identified from the literature related to green products. These keyword phrases were a combination of various adjectives (related to green), and nouns indicating different types (services and goods) and attributes (like label, brand, package, etc.) of a product offered by a marketer to the consumers. An effort was made to include all possible keywords phrases related to green products in the domain of Marketing Management. Further, to ensure the retrieval of all relevant records, various fields like the title of the publications, abstracts of the publications, and author-supplied keywords were searched for all the identified keyword phrases in the SCOPUS database. The SCOPUS database has been used as it is one of the most comprehensive data sources for literature analysis, and indexes over forty thousand journals, conferences, and book titles. The search query for the data collection was performed on 24 June 2020. A total of 23,794 records were extracted from the SCOPUS database against the keyword phrases. For some keywords, the multiplicity of occurrences are exactly the same, as the SCOPUS database may have used synonyms and whitespace in such cases and returned exactly the same records. The keyword phrase-wise number of documents available from the SCOPUS database are presented in Table 1.

In the next step, the duplicate records were removed, and a total of 15,678 records covering journals, books, and conferences were left. The number of records was further pruned based on language, document type, and research areas. For the purpose, all document types in the English language from the selected research areas (Business, Management, Accounting, Social Science, Economics, Econometrics and Finance, Arts and Humanities, Decision Science, Psychology, Multidisciplinary, and Neuroscience) were considered. Considering the relevance of green products to the domain of Marketing Management, the research areas having direct or indirect linkage with this domain were selected based on expert opinion. The rest of the research areas having very remote or no linkage with the domain of Marketing Management like Physics, Chemistry, Mathematics, Nursing, Fishery, Dentistry, Pharmacy, Agriculture, Crime Prevention, etc. were excluded from the results in consultation with the experts. This resulted in the selection of a total of 1720 records out of 15,678 records. To extract the records, a complex query with required considerations could have also been executed. However, to ensure the inclusion of all relevant publications, various records were first extracted against the keyword phrases relevant to the domain of Marketing Management and then pruned further. Thereafter, the titles, abstracts, and author-supplied keywords of these 1720 records were scrutinized for relevance against the 60 keyword phrases by experts consisting of a group of five academicians. These experts manually examined the abstracts of all the 1720 publications so that any publication not thoroughly covering the green products in the domain of Marketing Management could be excluded from the further analysis. The experts had at least 10 years of research experience in the fields of Marketing Management and/or Sustainability. Finally, a total of 1619 records were left for further analysis.

For each of the finally selected 1619 records, various pieces of information like publication title, author name(s) and affiliation, journal name, number, volume, pages, date of publication, abstract, cited references and author-supplied keywords were extracted from the SCOPUS database for bibliometric analysis. Thereafter, a bibliometric analysis was undertaken to ascertain the leading sources (journals), countries, affiliations, year, author, and publications in the field of green products. Further, to examine research growth in the area of green products and establish future research directions, co-occurrence analysis of author-supplied keywords was undertaken. For the purpose, visualization of the keyword terms in the field of green products was created by developing the network diagram using the VOSViewer software.

## 3. Bibliometric Analysis of Literature: Results

The publications concerning the green products and the related aspects indexed in the SCOPUS database were found distributed in thirteen document types. The most common format of publication was articles (72.41%), followed distantly by book chapters (12.16%), conference papers (4.44%), reviews (4.16%), and books (3.83%). Other forms of publication were less than one percent. It was further found that the records pertaining to green products during the chosen period of 1964–2019 were published in twenty different languages with English (96.79%) being the most dominant linguistic form of communication.

### 3.1. Publication Trends of Green Product Research

The journey of research in the field of green products and related areas is spread over a period of 56 years since 1964. Starting with two publications in the year 1964, the field related to green products saw 27 publications in 2005, 23 publications in 2006, and 185 publications in 2019. It is found that the initial period of 42 years from 1964–2005 has contributed only 11.12% of total publications. However, the last 14 years from 2006–2019 are found to be the most productive as they have contributed to 88.88% of the total publications during the study period of 1964–2019. There has been an exponential growth in the publications related to the area of green products since 2006 (y = 0.258 e^0.109x^, y = total publications, x = time in years, *r*^2^ = 0.8679) as reflected in Figure 2. An exponential curve is fitted on to the data in Figure 2 by using exponential regression. The relative predictive power of the exponential model is denoted by *r*^2^ also known as the coefficient of determination. The value of *r*^2^ (0.8679) indicates that 86.79% of the total variation in the number of publications is explained by the relationship between y (total publications) and x (time). Thus, the percentage growth of publications related to green products over time is expected to be 86.79%.

The number of publications is the key to establish the progress of any research field [52]. However, the citations scored by the article since its publication indicate the global impact of the publication [53]. The selected 1619 publications on green products accumulated a total of 24,447 citations with an average citation per paper (ACPP) of 15.10 citations. ACPP is calculated as total citations (TC) divided by total publications (TP). In terms of the year with the highest number of citations, it was found that the year 2010 had received the highest number of citations (3557) for the publications produced in this year with an average citation per paper of 50.10 citations. Out of 71 publications in 2010, four publications were highly cited with each having 400 or more citations. The citation data was taken until December 2019.

### 3.2. Country Productivity

The analysis of the 1619 records revealed that these publications were published from 72 countries. The top ten most productive countries with fifty or more publications in the field of green product research are presented in Table 2.

Table 2 reveals that among the most productive countries producing research related to the green product field, the leading country with a contribution of 418 (25.82%) publications out of a total of 1619 was the United States of America (USA). The USA is distantly followed by the United Kingdom (9.57%), India (7.16%), Australia (5.93%), Germany (4.94%), China (4.76%), Canada (3.95%), Taiwan (3.58%), Italy (3.40%) and Malaysia (3.40%), in that order. As far as total citations are concerned, the top-ranked country USA (7022) was distantly followed by Taiwan (1706), Canada (1369), Australia (1261), and the United Kingdom (1110), in that order. Taiwan (29.41) was found to occupy the top rank in the case of average citation per paper followed by Canada (21.39), the USA (16.80), Italy (16.00), and Australia (13.14), in that order. The average citation per paper can be used as a parameter of research valuation, and it is found that the publications from Taiwan and Canada are the most frequently cited as compared to those from the most productive countries. This is even though these countries have otherwise a far lower number of publications than the most productive countries.

The progression of research publications in the field of green products by the top five most productive countries is given in Figure 3.

It is found from Figure 3 that authors from the USA were publishing in the area of green products since the year 1969. The first publication from Germany appeared in 1978 and that from the United Kingdom appeared in 1982. Further, the authors from Australia were publishing since 1984. The publications in the area of green products from the Indian authors appeared only in 2003. However, in the last few years, India surpassed Germany and Australia in terms of the number of publications.

### 3.3. Productive Authors

This section discusses the author’s productivity in the area of green products. It is found that 1619 publications were contributed by 3699 authors either singly or in joint authorship. It is also revealed that 27 publications on green products were without author details. Further, out of the remaining 1592 publications, 505 were single-authored, whereas the rest of the 1087 publications were multi-authored, authored by a total of 2836 authors. The maximum number of authors who jointly co-authored one publication was reported as 36. It is also found that 482 authors have contributed only one publication in the field of green products. The authorship pattern for single or multi-authored 1592 publications is presented in Figure 4.

Figure 4 reveals that there was a large dispersion in the authorship pattern. In line with Lotka’s Law [54], it is found that not many authors were involved in producing a large number of publications. Most of the publications were authored by either two authors (31.93%) or a single author (31.19%). The domination of small teams of authors and quite a high percentage of single-authored publications as indicated by Figure 4 shows that Lotka’s Law holds in the field of research related to green products. The topmost productive authors in the field of green products are listed in Table 3. It is revealed that M. Charter was the most productive author with the highest number of publications (6), highest *h*-index (5), and the second rank in total citations (69). M. Charter authored his first article in the area of green products in the year 2008. However, out of the list of the most productive authors, J. Thøgersen started publishing before any other author in the year 2000. J. Thøgersen and N. Pandey jointly followed M. Charter with the second rank when it comes to the number of publications (5). The rest of the authors in the list of the most productive authors produced four publications each. However, J. Thøgersen with 542 citations has the top rank in total citations and is distantly followed by M. Charter (69), P. Castka, (66), A. Lobo and J.J. Zhang (62), U. Tischner (55), and P. Cozens (40), in that order. J. Thøgersen also shared the first rank with M. Charter as far as *h*-index (5) was concerned. N. Pandey who jointly held the second rank with J. Thøgersen in respect of the total publications started publishing in the year 2018 only and stood at twelfth rank in total citations.

### 3.4. Productive Journals

Table 4 presents a list of the most productive journals in the field of green products. The most productive journals have contributed 9.45% of the total 1619 publications.

From Table 4, it can be observed that the *Journal of Business Ethics* with the 2019 journal impact factor (IF_2019_) of 4.141 was the top-ranked journal in the field of green products, and it published in this area for the first time in the year 2004. It has published 37 articles related to green products with 2441 total citations. Further, it is revealed that the *Journal of Business Research* (IF_2019_ = 4.874) was the second most productive journal and it initiated publishing in the field of green products from the year 2000. This journal published 19 articles concerning green products with 975 total citations. In terms of the number of publications, the *Journal of Business Research* further followed *Quality Access to Success* (16), *Journal of Consumer Marketing* (15), and *International Journal of Sustainability in Higher Education* (14), in that order. Out of the list of the most productive journals, the most recent journal to publish on the green product area was *Industrial Marketing Management.* This journal started publishing in this area from the year 2017 only and has published 9 articles so far with 65 total citations. On the parameter of impact factor as a measure of journal evaluation [55], only four journals out of the most productive list of journals had impact factors in the year 2019. *Journal of Business Research* had the highest impact factor of 4.874, followed by *Industrial Marketing Management* (IF_2019_ = 4.695) and *Journal of Business Ethics* (IF_2019_ = 4.141), and *International Journal of Sustainability in Higher Education* (IF_2019_ = 2.000), in that order. The rest of the journals in the list of most productive journals were not indexed in the Journal Citation Report.

The yearly growth of top sources in the field of green products is presented in Figure 5. It is found that in recent years, *Quality Access to Success* and the *Journal of Business Ethics* published more articles than any other journal in the area of green products. In the year 2018, *Quality Access to Success* published maximum publications though this journal started publishing in the field of green products in the year 2013 only. On the other hand, the *Journal of International Consumer Marketing*, which first published on green products before any other journal in the year 1996, has produced very few articles in this area so far.

### 3.5. Productive Institutes

The most productive institutes in the field of green products are presented in Table 5. The top eight institutes have contributed 5.06% of the total 1619 publications. It is further revealed that the most productive institute was the University of California, Berkeley, with 12 publications in the field of green products, 650 total citations, an average citation per publication of 54.17, and an *h*-index of 7. Further, Aalto University, Finland, also produced 12 publications related to green products. However, this University could achieve only 84 total citations, an average citation per publication of 7.0, and an *h*-index of 6. Universiti Sains Malaysia, Malaysia, produced 11 publications in the area of green products. Further, Hong Kong Polytechnic University, Hong Kong, and Norges Teknisk-Naturvitenskapelige Universitet, Norway, produced 10 publications each, whereas Ohio State University, USA, University of Canterbury, New Zealand, and Bucharest University of Economic Studies, Romania, produced nine publications each. Hong Kong Polytechnic University, Hong Kong, stood at second rank after the University of California, Berkeley in respect of total citations (534) and average citation per publication (53.40). Table 5 also shows that Ohio State University, USA, with the highest *h*-index (9) followed the University of California, Berkeley, and Hong Kong Polytechnic University, Hong Kong, to rank at the third position in the case of total citations (458) and average citation per publication (50.89).

### 3.6. Most Cited Articles

The citedness of an article is a quantitative measure based on the number of citations accumulated by the articles since it is published. The most cited articles in the area of green products have been analyzed based on citations in the year of publication (TC_0_), citations in the year of study (TC_2019_), total citations (TC), and average citations per year (ACPY). Table 6 shows the top eight most cited articles on green products.

Table 6 reveals that the most cited article with a total of 481 citations was authored by Chen et al. [56]. It was the most impactful in terms of both early citations (TC_0_ = 4) as well as the number of citations in the year 2019 (TC_2019_ = 108). The second most cited article was authored by Daily and Huang (TC = 392) [57], followed by articles authored by Egri and Herman (TC = 383) [58], Tanner and Kast (TC = 380) [59], Dangelico and Pujari (TC = 364) [60], Hall et al. (TC = 363) [61], Luchs et al. (336) [62], and Chen (TC = 333) [63], in that order. Further, the articles authored by Tanner and Kast [59] and Luchs et al. [62] did not score any citations in the year of their publications.

Figure 6 indicates the citation life cycle of the highly cited articles in the area of green products. It is revealed that among the most cited articles, the earliest article published in the year 2000 was authored by Egri and Herman [58]. This article has scored 19.15 average citations per year. Figure 6 further shows a decline in the number of citations for this article since 2015. The citedness of the article indicates the impact of the research. Two different patterns of the citation life cycle of an article have been reported [64] as (i) an early rise in citations followed by a rapid decline and (ii) a delayed rise in citations followed by a delayed decline. The articles following the first pattern of the citation life cycle have a lower number of overall citations and the articles following the second pattern have a higher number of overall citations. The citation life cycle of the highly cited articles in the present study as indicated in Figure 6 seems to reflect almost both the patterns. It can be seen that the article authored by Egri and Herman (TC = 383, ACPY = 19.15) [58] shows a later decline in the early rise character of the citation pattern. However, the articles authored by Daily and Huang (TC = 439; ACPY = 20.63 citations) [57] and Tanner and Kast (TC = 419; ACPY = 22.35 citations) [59] show a delayed rise in the citation pattern. Further, Figure 6 indicates that all other highly cited articles have shown an early rise in the citation patterns. The decline in the citation patterns of these highly cited articles is yet to be observed.

### 3.7. Author Keywords Analysis

To understand the growth in an area of study, the concept of keyword extraction can be used [65]. The extant literature shows that many bibliometric studies [66,67] have employed keyword extraction to examine the growth of a subject area.

The network diagram is prepared by using VOSViewer software to create a visualization of the co-occurrence of the keyword terms in the domain of the subject area. The co-occurrence is computed as the number of times two keywords appear together in publications. In a network diagram, the keyword terms in different clusters are displayed using different colors. The keyword terms grouped into the same cluster are more likely to reflect identical topics. The keyword most in common is the largest node for that cluster [68,69]. Further, the changes in the colors of the cluster as one moves from one cluster to another reveal how the area of the study has progressed.

In the present study, the co-occurrence of keywords in author-supplied keywords has been examined. Figure 7 shows the visualization of the co-occurrence of author-supplied keywords in the form of the mean network diagram plotted by using VOSViewer. The minimum occurrence of the words plotted for the mean network diagram is five. Purple nodes correspond to the keywords used at the beginning of the study period, and red nodes correspond to the keywords that have appeared more recently. It can be found that Figure 7 has grouped the keyword terms into five major nodes, viz., sustainability, green marketing, sustainable development, sustainable design, and green products.

The purple-colored cluster with sustainable design being the largest node in Figure 7 indicates the theme areas used in the initial phase of the research concerning the green products. The mean value of co-occurrence of the keyword term of sustainable design is found to be close to 0.9990 with associated keyword terms being logistics, reverse logistics, green design, sustainable product development, innovation, benchmarking, marketing strategy, ethics, identity, etc. This indicates that in the initial phase of the research leading to the concept of green products, the researchers from diverse fields like technology and design, supply chain, marketing, human behavior, etc. worked on a wide number of theme areas. Further, only one country, that is, the USA, appears in the purple-colored keywords indicating the significant early contribution of the researchers from the USA in the area of green products. This coincides with what was revealed in the country productivity.

Figure 7 also shows that the concepts learned in this initial phase were further applied by researchers in the area of marketing. This is revealed by the keyword terms like green consumerism, new product development, eco-labeling, etc. that reflect light blue cluster with the largest node being the keyword term of green marketing. Further, the appearance of China in this cluster shows the interest of Chinese authors in this area. The mean value of the co-occurrence of the green marketing node with associated keywords is found to be in the range of 0.9990 to 0.9995. Thereafter, Figure 7 further indicates that the researchers attempted to relate the available theme areas to topics like corporate social responsibility, stakeholders, environmental consciousness and innovation, competitive advantage, etc. that belong to the indigo-colored cluster and converge into the largest node of sustainable development. It is revealed that the mean value of the co-occurrence of this sustainable development node with associated keywords is in the range of 0.9995 to 1.000. Thus, it seems that authors started relating sustainable development with stakeholder’s perspective and corporate responsibility.

It can be further found from Figure 7 that the available theme areas related to sustainable development were further adopted by researchers working in the field of marketing as shown by the green-colored cluster of keyword terms like marketing, green process innovation, green satisfaction, green brand image, etc. Thereafter, the progression to the green product field saw the application of the knowledge of various available topics by researchers to the theme areas like consumer behavior, green product innovation, green supply chain management, green management systems, etc. that belong to the lemon-colored cluster. The largest node of this lemon-colored cluster is the keyword term sustainability that has a mean value of close to 1.0005 for co-occurrence with the associated keyword terms. Hence, it can be assumed that the authors looked for linkage between marketing and sustainability to venture into the theme of the green products (orange-colored cluster) with related keyword themes like sustainable consumption, green innovation, brand equity, green supply chain, green brand, green advertising, etc. Figure 7 shows that the mean value of the co-occurrence of this green product node with the associated keywords is in the range of 1.005 to 1.010.

The orange-colored cluster of keyword terms with the green products as the node is followed by the red-colored cluster of keyword terms. Thus, keyword terms belonging to the red-colored cluster indicates the theme areas into which the green product research is advancing in recent times. Hence, it can be presumed that progression of research to green product area is currently advancing into the themes like greenwashing, green consumption, green purchase intention, willingness to purchase, environmental attitude and knowledge, brand loyalty, health, young consumers, etc. as these keyword terms belong to the red-colored cluster. The recent literature also recognizes that academic research concerning green products is advancing considerably in the areas of greenwashing [70,71], environmental knowledge [72], green consumption [73], green purchase intention [74,75], green branding and loyalty [76], health consciousness [77], etc. in recent times. Further, the young consumer also appears among the red-colored cluster of keywords indicating that authors are also exploring the linkage of green products with the strata of young consumers as confirmed by the available studies [78,79]. The above author keyword analysis shows that there has been a clear maturation in the field of research concerning green products from the early debates on sustainable design, green marketing, sustainable development, and sustainability. It is also to be noted that over the different phases of development of the research themes leading to green products and further, some of the themes have been repeatedly adapted as clear definitions to describe these topics do not exist.

## 4. Discussion

This study presents a bibliometric analysis of the literature on green products in the domain of Marketing Management between 1964 and 2019. The study is based on the 1619 publications concerning green products extracted from the SCOPUS database. The findings summarize publications in this domain by highlighting the salient features of published research like publication trends; authorship patterns; and leading publications, authors, journals, institutions, and countries.

It is found that research in the field of green products in the domain of Marketing Management spans over the last 56 years. From a slow beginning, the trajectory has been exponential after the year 2006 with the period 2006–2019 being the most productive. The percentage growth of publications related to green products over time is expected to be 86.79%. This exponential growth is not limited to the increase in output in this area of research by one specific country or journal. The field has received research contributions from as many as 72 countries. It is further found that the maximum number of global publications has been contributed by the USA. The USA also has the top rank in total citations. However, the average citation per publication indicates that publications from Taiwan and Canada are the most frequently cited. This is despite these countries having otherwise a far lower number of publications than the most productive countries. In the current phase, authors from the United Kingdom and India are also contributing significantly to this area of research. Further, some high-impact journals have contributed to the growth of the research in the green product area. *Journal of Business Ethics* with an impact factor of 4.141 was the top-ranked journal and published a total of 37 articles with 2441 total citations. It is found that most productive journals cumulate about 9.45% of the total 1619 publications.

The findings further indicate that most of the publications were authored by either two authors (31.93%) or a single author (31.19%) indicating that Lotka’s Law holds in the field of research related to green products. Thus, it can be presumed that there is less tendency on the part of the authors to team up, and the majority of them like to work in isolation. The findings also reveal that M. Charter is the top author with the highest number of publications in the area of green product research. However, the most cited article is by Chen et al. [56]. Further, the most productive institute in the field of green products is the University of California, Berkeley. It is also found that the most productive institutions contributed about 5.06% of the total 1619 publications related to green products in the domain of Marketing Management. The findings with respect to the publication pattern, citations, and influencing authors and other entities on the green product related field give an opportunity to appreciate the evolution of the field and inform about contributions of various actors in the field.

The present study also offers insights into the evolution and growth of research leading to the field of green products. For the purpose, a network diagram indicating the co-occurrence of the author-supplied keywords has been presented. This network diagram has been able to showcase the broad theme areas like sustainable design, green marketing, sustainable development, and sustainability that have finally led to the interest of the researchers in the field of green products. Using this diagram, the study has also been able to identify the theme areas into which green product research is advancing in recent times. These theme areas include greenwashing, green consumption, green purchase intention, willingness to purchase, environmental attitude and knowledge, brand loyalty, health, young consumers, etc. Thus, the study provides useful information and research trends with regards to the past, present, and future of the green product field. The study offers a guide to the researchers who wish to pursue research in this area.

The findings also show that research on green products is relatively recent and has its roots in the framework of the literature that is entrenched broadly in the fields of technology, supply chain, sustainability, and marketing. It can be presumed that the topic of green products is of current relevance for the researchers. The topic seems to be advancing into a variety of green themes related to consumer trust and purchase intentions, branding and loyalty, and environmental and health consciousness. It seems that researchers wishing to undertake studies in this area would have an exciting journey ahead as there is still much to discern.

## 5. Conclusions, Limitations, and Directions for Future Research

The research field of green products in the domain of Marketing Management has experienced significant growth since its evolution in 1964. This growth has been exponential, especially in the last 14 years period of 2006–2019. The increasing research contributions have reflected a noteworthy impact on the availability of literature in the field of green products. The study has further revealed publication and citation growth patterns, and leading publication sources, authors, institutions, and countries; thus, offering potential implications to the researchers and practitioners. The availability of data on top authors is expected to guide the researchers seeking studies in green products to achieve research networking. The most cited articles are also indicated in the study. The study further reveals that the most productive journals cumulate 9.45% of green products related publications. This offers an opportunity for potential researchers to minimize their efforts in accessing avenues where most of the green product-specific research has been published. Further, the most productive institutions have been found to contribute about 5.06% of the total number of publications. This data presents ready-hand information to the potential researchers and marketers about institutions specializing in this field. This information could be useful in seeking collaborations, research guidance and support, and expert opinion on green product modification or development; thus, contributing to the growth of the field of green products. Further, the findings with respect to the network analysis of the co-occurrence of the author-supplied keywords concerning the green products offer insights for the potential researchers to identify the evolution and growth of the topic. Thus, they may ascertain current and potential areas of research in the field of green products and contribute to the augmentation of the field. In conclusion, it can be presumed that green product makes up a topic that has been recently developed and currently entails great relevance both for academics and practitioners. This might be explained by the number of scholars from diverse fields like technology and design, supply chain, marketing, sustainability, human behavior, etc. who have ventured to research this topic, which in turn indicates the strong research interest that this topic has attained over time.

The results presented and discussed in the study are subject to a few limitations. Firstly, the present study is based on a sample of records available in the SCOPUS database. There may be many studies on the green products that are published in various other journals, not accessible through the SCOPUS database. Secondly, an effort was made to include all possible keywords phrases and research areas relevant to green products in the domain of Marketing Management. However, there could be a possibility of omitting a related keyword or research area. Thus, the likelihood of sampling error with respect to the extracted publications on green products is not ruled out. Thirdly, while using bibliometric tools to analyze data, some extracted studies from the database have different authors with the same names. In the present study, all the records were scrutinized to avoid such errors. However, there could be a chance of a few such studies having been considered. Fourthly, different terms like sustainable product, environmental product, ecological product, and eco-product are used interchangeably in the literature to describe green products resulting in overlapping definitions and concepts. Fifthly, it should be noted that the findings of the present study only offer a snapshot of the currently available research in the area of green products. However, since research in this area is still evolving, the data with respect to the publication trends; authorship pattern; and leading publications, authors, journals, institutions, and countries may change over time. Finally, since the present study has examined research related to a specific area of green products relevant to the domain of Marketing Management, future researchers have to be cautious while generalizing these results to researches spanning across various other green practices and processes.

To present a more comprehensive understanding of the topic, future bibliometric studies in the area of green products may examine various non-indexed journals and other available databases like Web of Science, Google Scholar, EBSCOhost, etc. The researchers in the future may achieve better results by comparing various interchangeably used terms like “green products”, “sustainable products”, “environmental products”, “ecological products”, and “eco-products”, and analyzing extracted data against each of these terms individually. Future research may also undertake a co-citation analysis along with other bibliometric parameters not covered in the present study. They can further utilize a structural indicator and sociogram to examine the associations between publications, authors, journals, institutions, and countries.

## Figures and Tables

**Figure 1 ijerph-17-08469-f001:**
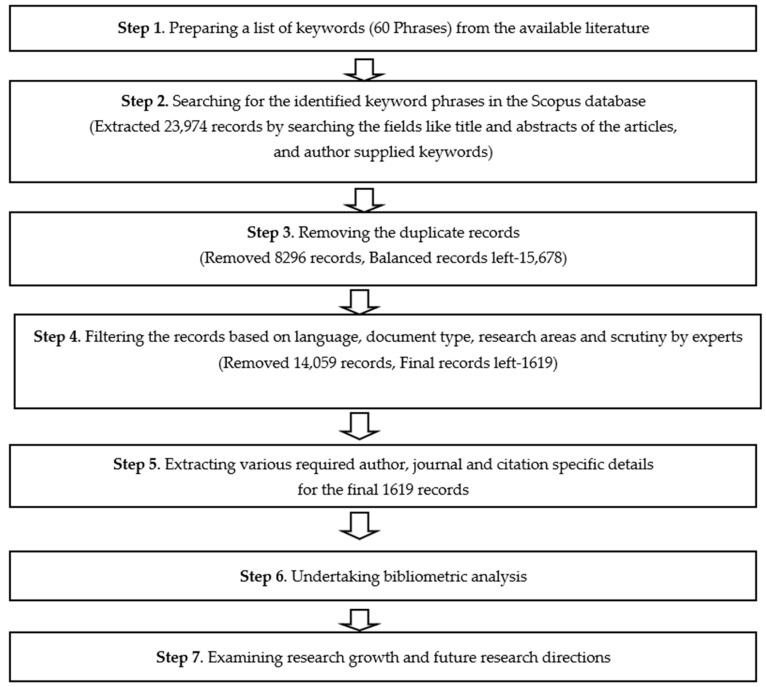
Data collection and analysis approach.

**Figure 2 ijerph-17-08469-f002:**
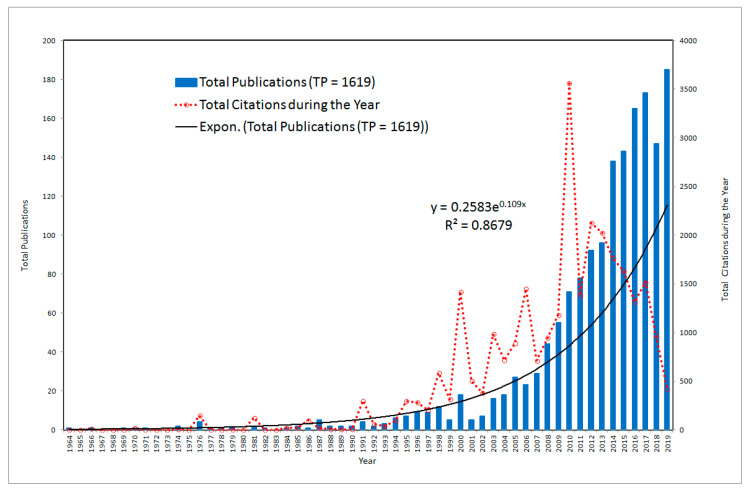
Year-wise research production and their total citation.

**Figure 3 ijerph-17-08469-f003:**
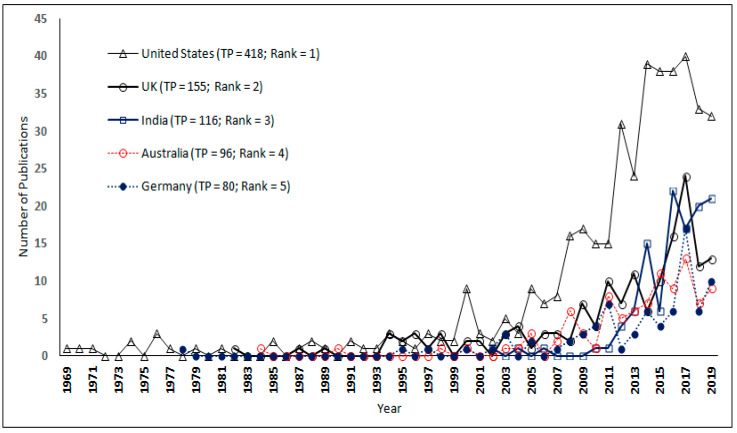
Progression of the research publications by the top five countries.

**Figure 4 ijerph-17-08469-f004:**
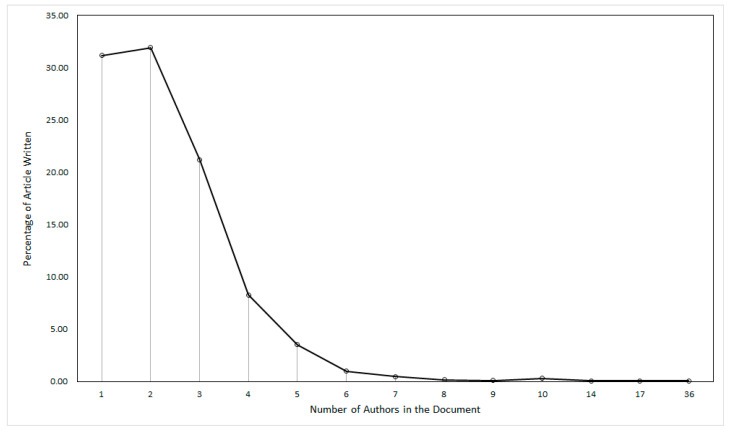
Authorship pattern of green product research.

**Figure 5 ijerph-17-08469-f005:**
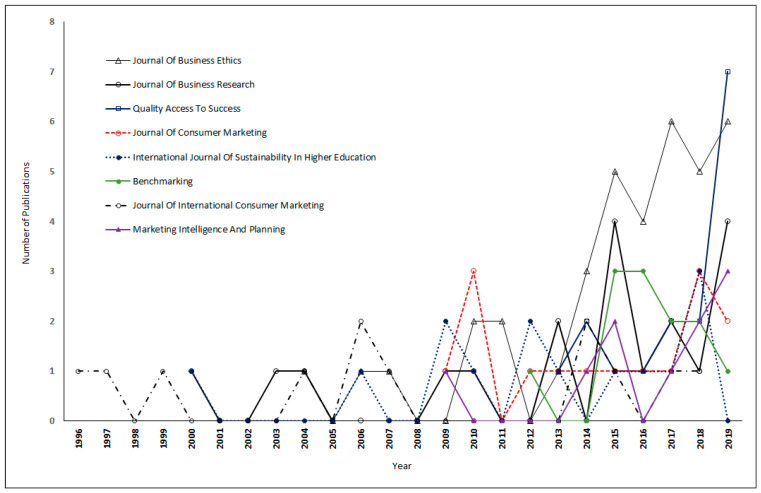
Yearly growth of top sources in the field of green products.

**Figure 6 ijerph-17-08469-f006:**
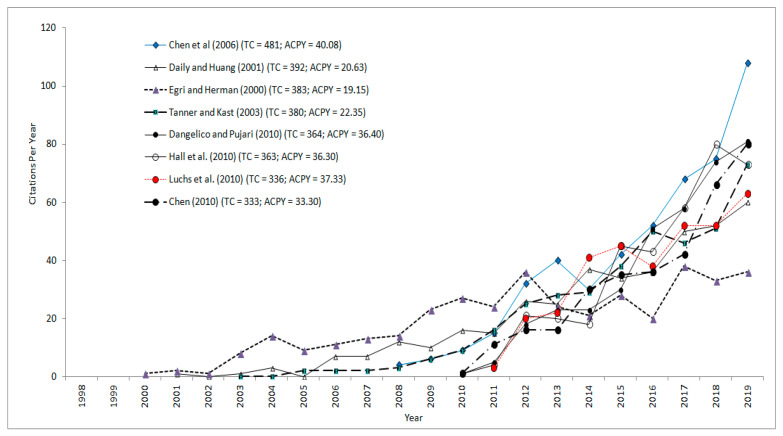
The citation life cycle of the highly cited articles in green products. (TC–Total Citations, ACPY–Average Citations per Year).

**Figure 7 ijerph-17-08469-f007:**
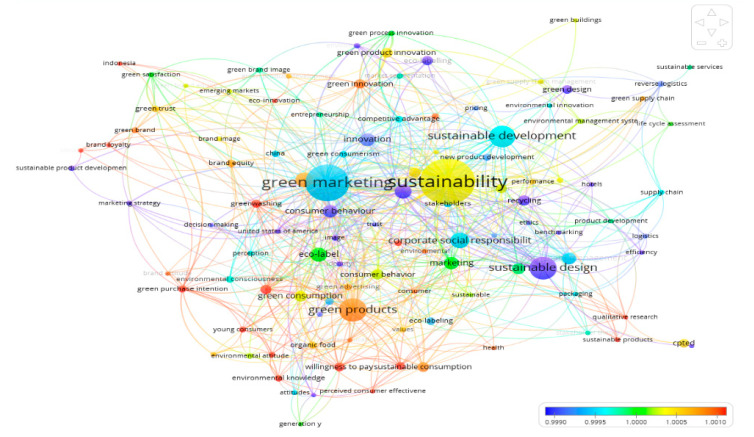
Network diagram based on the mean value of co-occurrence of keywords.

**Table 1 ijerph-17-08469-t001:** Keyword phrases related to the green product.

Keyword	Response from SCOPUS	Keyword	Response from SCOPUS	Keyword	Response from SCOPUS
“Green Product”	2834	“Sustainable Brand”	70	“Environmental Design”	4305
“Green Service”	148	“Sustainable Branding”	9	“Environmental Symbol”	13
“Green Good”	57	“Sustainable Package”	22	“Environmental Logo”	3
“Green Offering”	13	“Sustainable Packaging”	376	“Environmental Signage”	2
“Green Brand”	160	“Sustainable Label”	22	“Environmental Signboard”	0
“Green Branding”	45	“Sustainable Labelling”	10	“Eco-Brand”	17
“Green Package”	53	“Sustainable Labeling”	10	“Eco-Label”	812
“Green Packaging”	268	“Sustainable Design”	4007	“Eco-Branding”	19
“Green Label”	171	“Sustainable Symbol”	0	“Eco-Labelling”	703
“Green Labelling”	61	“Sustainable Logo”	1	“Eco-Labeling”	703
“Green Labeling”	61	“Sustainable Signage”	0	“Eco Brand”	17
“Green Design”	1331	“Sustainable Signboard”	0	“Eco Label”	812
“Green Symbol”	11	“Environmental Product”	776	“Eco Branding”	19
“Green Logo”	5	“Environmental Brand”	14	“Eco Labelling”	703
“Green Signage”	1	“Environmental Branding”	2	“Eco Labeling”	703
“Green Signboard”	0	“Environmental Package”	26	“Renewable Product”	148
“Sustainable Product”	2174	“Environmental Packaging”	42	“Renewed Product”	9
“Sustainable Service”	521	“Environmental Label”	170	“Recyclable Product”	131
“Sustainable Good”	87	“Environmental Labelling”	220	“Recycled Product”	659
“Sustainable Offering”	16	“Environmental Labeling”	220	“Recycled Offering”	2

**Table 2 ijerph-17-08469-t002:** Top ten most productive countries in the field of Green product research.

Country	TP	R (%TP)	R (TC)	R (ACPP)
United States	418	1 (25.82)	1 (7022)	3 (16.80)
United Kingdom	155	2 (9.57)	5 (1110)	7 (7.16)
India	116	3 (7.16)	8 (578)	9 (4.98)
Australia	96	4 (5.93)	4 (1261)	5 (13.14)
Germany	80	5 (4.94)	7 (844)	6 (10.55)
China	77	6 (4.76)	23 (191)	10 (2.48)
Canada	64	7 (3.95)	3 (1369)	2 (21.39)
Taiwan	58	8 (3.58)	2 (1706)	1 (29.41)
Italy	55	9 (3.40)	6 (880)	4 (16.00)
Malaysia	55	9 (3.40)	17 (312)	8 (5.67)

(TP: Total Publications; R: Rank; TC: Total Citations; ACPP: Average Citation Per Paper).

**Table 3 ijerph-17-08469-t003:** Most productive authors in the field of green products.

Author	Affiliation	(TP)	R (TC)	*h*-Index	PY_Start
M Charter	University for the Creative Arts (UCA), London, United Kingdom	6	2 (69)	5	2008
J. Thøgersen	Aarhus Universitet, Aarhus, Denmark	5	1 (542)	5	2000
N. Pandey	National Institute of Industrial Engineering, Mumbai, India	5	12 (18)	2	2018
P. Castka	University of Canterbury, Christchurch, New Zealand	4	3 (66)	3	2016
A. Lobo	Swinburne University of Technology, Melbourne, Australia	4	4 (62)	3	2017
J.J. Zhang	University of Victoria, Victoria, Canada	4	5 (62)	3	2012
U. Tischner	Ec[o]ncept, Germany	4	6 (55)	3	2017
P. Cozens	Curtin University, Perth, Australia	4	7 (40)	3	2016
S. Kajalo	Aalto University, Espoo, Finland	4	8 (36)	4	2010
A. Lindblom	Aalto University, Espoo, Finland	4	9 (36)	4	2016
H.J. Wang	Fo Guang University, Jiaosi, Taiwan	4	10 (26)	3	2016
F. Rubik	Institut für Ökologische Wirtschaftsforschung, Berlin, Germany	4	11 (21)	3	2008
V. Sima	Universitatea Petrol-Gaze din Ploiesti, Ploiesti	4	13 (10)	1	2009

(TP–Total Publications, TC–Total Citations, *h*-index-Hirsch Index, PY_Start–Publication Starting Year).

**Table 4 ijerph-17-08469-t004:** Most productive journals in the field of green products.

Journal	TP	TC	IF_2019_	PY_Start
Journal of Business Ethics	37	2455	4.141	2004
Journal of Business Research	19	975	4.874	2000
Quality Access to Success	16	41	-	2013
Journal of Consumer Marketing	15	798	-	2009
International Journal of Sustainability in Higher Education	14	237	2.000	2000
Benchmarking	12	108	-	2012
Journal of International Consumer Marketing	12	559	-	1996
Marketing Intelligence and Planning	10	230	-	2009
Industrial Marketing Management	9	65	4.695	2017
Journal of Consumer Policy	9	10	-	2017

(TP–Total Publications, TC–Total Citations, IF_2019_–2019 Journal Impact Factor, PY start–Publication Starting Year).

**Table 5 ijerph-17-08469-t005:** Most productive institutes in the field of green products.

Affiliation	TP	TC	ACPP	*h*-Index
University of California, Berkeley, USA	12	650	54.17	7
Aalto University, Finland	12	84	7.00	6
Universiti Sains Malaysia, Malaysia	11	136	12.36	5
Hong Kong Polytechnic University, Hong Kong	10	534	53.40	8
Norges Teknisk-Naturvitenskapelige Universitet, Norway	10	30	3.00	4
The Ohio State University, USA	9	458	50.89	9
University of Canterbury, New Zealand	9	81	9.00	2
Bucharest University of Economic Studies, Romania	9	13	1.44	2

(TP–Total Publications, TC–Total Citations, ACPP–Average Citation per Publications, *h*-index–Hirsch Index).

**Table 6 ijerph-17-08469-t006:** Topmost productive articles in the field of green products.

Authors	Article	TC_0_	TC_2019_	TC
Chen et al. (2006) [56]	The influence of green innovation performance on corporate advantage in Taiwan, *Journal of Business Ethics*. 67(4), 331–339	4	108	481
Daily and Huang (2001) [57]	Achieving sustainability through attention to human resource factors in environmental management, *International Journal of Operations and Production Management*. 21(12), 1539–1552	1	60	392
Egri and Herman (2000) [58]	Leadership in the North American environmental sector: Values, leadership styles, and contexts of environmental leaders and their organizations, *Academy of Management Journal*. 43(4), 571–604	1	36	383
Tanner and Kast (2003) [59]	Promoting Sustainable Consumption: Determinants of Green Purchases by Swiss Consumers, *Psychology and Marketing*. 20(10), 883–902	0	73	380
Dangelico and Pujari (2010) [60]	Mainstreaming green product innovation: Why and how companies integrate environmental sustainability, *Journal of Business Ethics*. 95(3), 471–486	1	81	364
Hall et al. (2010) [61]	Sustainable development and entrepreneurship: Past contributions and future directions, *Journal of Business Venturing*. 25(5), 439–448	1	73	363
Luchs et al. (2010) [62]	The sustainability liability: Potential negative effects of ethicality on product preference, *Journal of Marketing*. 4(5), 18–31	0	63	336
Chen (2010) [63]	The drivers of green brand equity: Green brand image, green satisfaction, and green trust, *Journal of Business Ethics*. 93(2), 307–319	1	80	333

(TC0–Citations in the year of Publication; TC2019–Citations in the year 2019; TC–Total citations).

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
