# Peer review of "Research Trends in Green Product for Environment: A Bibliometric Perspective"

_ijerph, 2020, doi:10.3390/ijerph17228469_

Round 1
Reviewer 1 Report
Bhardwaj et al present a paper on the bibliometric analysis of terms related to
"green" products. The authors curated a set of 60 keyword combinations which
were used to search Scopus with the resultant list of publications filtered and
analysed.
General Comments
Overall the manuscript is clearly written and presented, however, the authors
do need to make clear from the start what they mean by "green". As they mention throughout the paper environmentally relevant terms are regularly used interchangeably leading to confusion. The authors need to be careful not to add to the uncertainty. It is a colloquialism which needs a term of reference
particularly in the abstract: it is the first word used without clarification.
Please clarify. The authors also use the term "the green product" as a singular
entity throughout where it appears they mean the plural "green products". Please correct.
The results detailed and precise with a relevant discussion and thoughtful conclusions. Environmental research is not my area of expertise, however, it is
the opinion of this reviewer that this works adds to the field.
Specific Comments
Methodology
It is unclear in the methodology how the authors filtered the set of 15,678
records down to only 1,720 (p4 line 140) that's an almost 90% reduction. How can they be sure that their subset is representative of the whole? A specific set of research areas were selected by the authors. Why where they chosen and what where the dominant areas that were filtered out? If the severe reduction was an exercise in getting a more manageable dataset, then perhaps a random sampling would be a less selective method.
In the their analyses the author fit an exponential curve onto the data (p5 line
191 and Figure 2), but do not mention what method was used to fit it nor
calculate the r-squared. An exponential curve implies a percentage growth of the measured variable over time, it would be meaningful to the reader for the
authors to explain what that was for the green product literature.
Citations are an important metric for assessing impact of publications, but
they're not a good measure of "quality" (p5 193). There are many reasons why an is referenced and often not because of quality. The authors do not make clear what they mean by average citations per paper (ACPP) as sometimes it is the mean of total citations (TC) divided by total papers (TP) (e.g. Table 5), but most often it is not. For example, the overall ACPP is reported as 17.62 (p5 line 195) which is not the result of 24,447/1,619 = 15.10. The distribution of citations are highly skewed as most publications are never cited and a small number of very highly cited so a mean is not an appropriate summary statistic (see https://doi.org/10.1101/062109). Using the median will be less sensitive to the skewness in the citation data. With this in mind the spike in citations for 2010 (p6 line 197 and Figure 2) is interesting. Is it due to a small number of highly cited papers or a more general increase in papers cited? Is there a reason for the additional citation spikes in 2000 and 2006? Could it be they coincide with the hugely influential IPCC reports of 2001 and 2007?
Results
In Table 2 the single country publication (SCP) and internationally collaborated
paper (ICP) numbers would be better represented as a proportion of total papers for the country i.e. US could be 52% relative SCP (rank 5) and UK would be 42% (rank 8). Given that the US has published the most papers, by a long way, it is unsurprising it is also top for SCP and ICP a relative measure would be more interesting to the reader. Again, please check the ACPP values. Finally, Italy and Malaysia should be ranked equal 9th as they have the same number of publications, not 9th and 10th.
On p7 line 220 the authors mention that Taiwan and Canada are the top ACPP
countries despite having for fewer papers. Do the authors have any suggestion
as to why that may be?
In Figure 4, page 8, the plot of percentage of articles vs number of authors in the document there are appears to be an approx. 1% of articles with 0 authors. How is that possible?
In Table 3, page 8, I believe the affiliation of "M Charter" is incorrect. There is no "M Charter" at UCL. Martin Charter is at the University for the Creative Arts
(UCA), Farnham https://www.uca.ac.uk/staff-profiles/professor-martin-charter/.
In Table 4, page 9, several journals have an IF of "0.000" to indicate that they
have none. It would be clearer to put "NA" or "-".
There is a typo on p 9, line 268. "IF2019 = 4874" should be "IF2019 = 4.874".
Figure 5, page 10, represents the number of publications by journal over time,
however, joining the points with curves is meaningless and confusing. A stacked bar plot would be much clearer.
In section 3.6 (p10, lines 292-306) there are many references made to ACPP as a percentage. What does that mean? Or is that an error? On line 296 the ACPP for Aalto University is cited as 12.36 when it should be 7.0. There is also reference made to institutions' h-index (in Table 5). How is that calculated as it is usually an author-specific metric?
Reviewer 2 Report
Congratulations to the authors on the work done.
I just have two requests that I would like to see introduced in your article:
It would be interesting if the authors could include the analysis of co-citations, demonstrating if there is any pattern in the co-authorship clusters.
Regarding the analysis of keywords, I ask you to indicate the minimum occurrence of the words plotted for the network diagram (referring to figure 7).
Best Regards,
Reviewer 3 Report
This is a sound and worthwhile bibliometric analysis of over time in green products. The methods are appropriate and reasonably well described and applied. What’s missing is the discussion: now this section only summarizes the results. Compare your main results with published data. Consider any findings that run contrary to your point of view.
Specific remarks:
• Explain why you chose Scopus as a data source
• Explain why VOSViewer, there are also other tools, eg. Gephi
• Usually, bibliometric analyses are based on one type of document. Explain why books and conferences are included. I didn’t find any descriptive statistics about conferences and books.
• Neuroscience? Why? But this may be a lack of mine knowledge.
• Can you say something more about records that were excluded from the analysis by five experts?
• Fig. 1 – shouldn’t be there also a step with data cleaning?
• The division into pre-mature phase and mature phase has quantitative character, it should be underlined
• How did you get a number of 3699 authors? Was there any cleaning? Any specific method was applied?
• Explanation fig. 4 by Lotka’s law is wrong. It is not Lotka’s law. Fig. 4 shows the domination of small teams of authors and quite a high percentage of single authored papers. What does it mean for this field? Are there any reasons for such a pattern?
• Top authors in the field – any practical meaning?
• Table 3 – the most productive Journals cumulate about 10% of all publications. What does it mean for researchers who are seeking studies on green products? the most productive institutions provided about 5% of the total number of publications. What does it mean for this field?
• Emphasize your major conclusions and the practical significance of your study
Reviewer 4 Report
- In times of more and more widespread access to the Internet and the pursuit in scientific research to the possibility of repeating the experiment by other researchers to achieve transparency, I encourage you to publish the data (at various stages of their processing/cleaning and at the last stage, which allowed the data visualization in the article to be obtained) as open research data on the Zenodo or Figshare platform and link them in this article. The authors declare the use of Excel, so the data could be CSV or XLSX.
- A large part of the Figs. are histograms (with discrete values on the x-axis) - the authors connect individual points either with broken lines (Fig. 2 [citations], 3, 4, 6) or even! they interpolate them (Fig. 5), which may mislead the Reader. I know you can see this approach in other articles, but you should follow InfoVis developments and not stick to bad practices. Try a different presentation that is sufficiently visual but not misleading. For example, use the possibility of publishing in color.
- [50] "In recent times, the focus of sustainability-oriented firms has shifted from the adoption of clean technologies to producing environment-friendly green products(...)" - it is debatable whether, given the current pace of scientific development, the last 15 years can be called "recent times" - please put it differently
- [116] "The application of this software offers timeline visualization of citations, information about most-cited articles and indicates the subsequent impact of those citations" - It's a pity that these possibilities have not been used, but please note then that only the basic capabilities of the VOSViewer application were used
- [129] "The search query for the data collection was performed on June 24, 2020" - please disclose this query to the Readers - you can use a variable instead of specific keywords
- [125] "A list of 60 keyword phrases (shown in Table 1) relevant to the domain of green product was first identified from the literature related to the green product" - seems that these are combinations of adjectives (green, sustainable...) and nouns (product, good), but some of them were excluded (e.g. eco design) - explain it to Readers why, please.
- For some keywords, the multiplicity of occurrences are exactly the same (eco {-} label {l} ing) 703 - maybe it is worth explaining to Readers that the Scopus database used synonyms and whitespace here and returned exactly the same records?
- [136] "The number of records was further pruned based on language, document type, and research areas..." - What was the purpose of extracting all records from the database according to the given keywords, and then limiting them to the English language, doc type, etc? Since it was possible to execute a complex query that would filter both by language and by keywords? Please explain this to the Reader.
- [141] "Thereafter, the titles, abstracts, and author-supplied keywords of these 1720 records were scrutinized for relevance against the 60 keyword phrases by a group of five academicians." - Does this mean that more people contributed to this article than those mentioned as authors?
- [145] "For each of the finally selected 1619 records, various pieces of information like article title, author name(s) and affiliation, journal name, number, volume, pages, date of publication, abstract, cited, references and author-supplied keywords were extracted for bibliometric analysis." - extracted from where? Please explain this to the Reader.
- [176] "The most common format of publication was articles (72.41%), followed distantly by reviews (12.16%), reviews (4.44%), conference papers (4.16%), and books (3.83%)." - you have duplicated reviews word in this sentence.
- [176] "The most common format of publication was articles (72.41%), followed distantly by reviews (12.16%), reviews (4.44%), conference papers (4.16%), and books (3.83%)." - The fact that more articles are published than other publishing forms is nothing very revealing. How is it when it comes to relative values - e.g. of all works published in the areas selected by you (Business, Management, Accounting; Social Science, Economics, Econometrics and Finance, Arts and Humanities, Decision Science, Psychology, Multidisciplinary, Neuroscience) in individual groups of publication types.
- [191] "the publications related to this area since 2006 (y= 0.258 e0.109x and r2 = 0.8679)" - this exponent formula is illegible - in the tex, you can include full-fledged mathematical formulas.
- [193] "to adjudge the quality of a publication, it is imperative to ascertain the citations scored by the article" - according to your statement and looking at Fig.2 we could conclude that since 2010 the quality of articles dropped dramatically - maybe it is worth commenting on it so that the inattentive Reader does not draw wrong conclusions?
- [189] "Hence, the present study categorizes this quadrant of 2006-19 as the maturity phase." - Looking at Fig. 2, however, I can still see an upward trend - so maybe consider a different term than the mature stage, because in 10 years it may turn out that this is not true and that the huge increase took place in 2030? Scientists should be critical and forward-looking.
- [205] Table 2 - SCP and ICP do not sum up to TP. Please explain to the Reader what counting methodology you have adopted and why it does not sum up.
- [312] "product have been analyzed based on citations in the year of publication (TC0)" - does this mean that you compare the number of citations of an article that, for example, was published in January, to an article that was published in December of the same year? But maybe it's better to compare the cumulative number of citations of the year following the publication of a given article? Then the disproportions resulting from other months of publication should be smaller.
- [363] "The purple-coloured cluster with sustainable deign being the largest node" - deing being?
- [476] "This has been a serious problem that could have resulted in overlapping..." - this wording is too general and incomprehensible, please be more precise.
- [174] Subsection 3.1 and 3.2 have the same title - probably 3.1 should have a different title.
- To be precise, one should use a different identification of the article than the names of the authors (e.g. Fig. 6 - Chen et al) - the figures and tables should describe the research results on their own, and not expose the Reader to misinterpretation (the point is that the team of Chen et al. al could have written many different articles).
Round 2
Reviewer 4 Report
Most of the comments have been sufficiently corrected - although there is some concern about the solution (or rather masking) of the problem previously reported as note no. 16 (the same applies to the solution of note no. 6). This is not good enough proof of the authors' scientific reliability and proves that the presented results may be biased.
However, unfortunately, note no. 2 was overlooked or ignored by the Authors. I cannot approve a scientific article where the figure shows a negative number of publications (Figure 5. - e.g. values between 2000 and 2001). So once again, Figures 3, 4, 5, 6 are de facto histograms, and connecting individual points with lines makes no sense. But if the Authors insist on such a presentation of data, let they will consistently apply the same polylines for Figure 5 as for Figure 3. There is no reason to use an even worse and more misleading form of presentation for Figure 5 than that used for Figure 3.
